# Highly efficient blue organic light-emitting diodes based on carbene-metal-amides

Patrick J. Conaghan [1,3], Campbell S. B. Matthews [1,3], Florian Chotard [2], Saul T. E. Jones [1], Neil C. Greenham [1], Manfred Bochmann [2], Dan Credgington [1✉] & Alexander S. Romanov [2✉]

Carbene-metal-amides are soluble and thermally stable materials which have recently emerged as emitters in high-performance organic light-emitting diodes. Here we synthesise carbene-metal-amide photoemitters with CF$_3$-substituted ligands to show sky-blue to deep-blue photoluminescence from charge-transfer excited states. We demonstrate that the emission colour can be adjusted from blue to yellow and observe that the relative energies of charge transfer and locally excited triplet states influence the performance of the deep-blue emission. High thermal stability and insensitivity to aggregation-induced luminescence quenching allow us to fabricate organic light-emitting diodes in both host-free and host-guest architectures. We report blue devices with a peak external quantum efficiency of 17.3% in a host-free emitting layer and 20.9% in a polar host. Our findings inform the molecular design of the next generation of stable blue carbene-metal-amide emitters.

[1] Cavendish Laboratory, Department of Physics, University of Cambridge, J J Thomson Avenue, CB3 0HE Cambridge, UK. [2] School of Chemistry, University of East Anglia, Earlham Road, Norwich NR4 7TJ, UK. [3] These authors contributed equally: Patrick J. Conaghan, Campbell S. B. Matthews. ✉email: dan.credgington@gmail.com; a.romanov@uea.ac.uk

The development of organic light-emitting diodes (OLEDs) has continued for over 30 years since the discovery of electroluminescence from the fluorescent green emitter tris (8-hydroxyquinolinato)aluminium ($Alq_3$)[1]. An OLED emits light via radiative recombination of strongly bound excitons formed from electrically injected charge pairs. For injected charges with uncorrelated spins, excitons are generated in a 3:1 ratio of spin-1 triplet (T) and spin-0 singlet (S) states. In the absence of spin–orbit coupling, only the singlets can relax to the ground state via photon emission, and the energy of triplets is wasted[2]. Luminescent harvesting of triplet states can be achieved in several ways. Use of heavy metal complexes with high spin–orbit coupling, notably of iridium(III) and platinum(II), enables photo-emission via phosphorescence on the microsecond timescale[3–7]. Alternatively, compounds can be designed where the lowest-energy excited states, $S_1$ and $T_1$, are sufficiently close in energy for the thermal equilibrium between the two at typical operating temperatures to enable emission from the $S_1$ state via (reverse) intersystem crossing. This process is termed E-type (or thermally activated) delayed fluorescence (TADF)[8–10]. The development of materials for blue OLEDs, however, remains particularly challenging, with high-energy bimolecular interactions involving long-lived triplet excitons implicated as one of the primary limits to operational lifetime[11].

We have shown recently that linear, two-coordinated coinage metal complexes of the type (L)MX (M = Cu, Ag, Au) can show efficient photoluminescence (PL) via a delayed fluorescence mechanism, provided the ligand L is a strongly bound carbene capable of acting as π-acceptor and X is an electron-rich anion capable of acting as electron donor upon excitation[12–16]. Combinations of L = cyclic (alkyl)(amino)carbene (CAAC)[17,18] and X = carbazolate (Cz) have proved particularly successful and have become known as "carbene–metal–amide" (CMA)-type photoemitters[14]. These complexes are soluble in most organic solvents and sufficiently thermally stable to enable the fabrication of OLED devices by both solution processing and thermal vapour deposition techniques[13,14,19,20]. CMAs do not suffer from strong concentration quenching in the solid state, attributed to the lack of close metal–metal contacts, enabling the realization of host-free green devices with an external quantum efficiency (EQE or $\eta_{EQE}$) of 23.1%[19]. Due to the linear two-coordinate geometry, CMAs are conformationally flexible, with a low barrier for rotation about the metal-N σ-bond. The highest occupied molecular orbital (HOMO) is centred on the amide ligand, while the lowest unoccupied molecular orbital (LUMO) consists mainly of the p-orbital of the carbene C atom, with only a small metal

contribution to both. Excitation of the molecule thus leads to a charge-transfer (CT) type excited state. Both frontier orbitals are spatially well separated, commensurate with a small $\Delta E(S_1 - T_1)$ energy gap for CT excitations. This arrangement enables luminescence quantum efficiencies to approach 100% coupled with short, sub-microsecond excitation lifetimes for triplet states. The emission process involved has been subjected to a number of theoretical and spectroscopic investigations[21–23]; modelling has shown that, upon rotation about the metal-N σ-bond, the $\Delta E(S_1 - T_1)$ energy gap decreases and at high twist angles may approach zero[13,20–23].

As a consequence of the high polarity of CMA compounds, emission energies are sensitive to their molecular environment; for example, this has allowed "tuning" of electroluminescence by suitable host media from green to sky-blue[19]. However, much larger changes in emission energies can be achieved by altering the carbazole substitution pattern. For the ($^{Ad}$CAAC)AuCz archetype, the HOMO is almost entirely located on the Cz donor. The introduction of electron-withdrawing groups to this moiety therefore influences the HOMO more than the LUMO and widens the HOMO–LUMO gap, resulting in a shift towards blue emission[14,24].

Here we report the synthesis of new Au-bridged emitters that enable blue host-free OLEDs with $\eta_{EQE}$ of 17.3% ($\lambda_{em,max} = 473$ nm), as well as host-guest devices with a peak wavelength of 450 nm and an $\eta_{EQE}$ of up to 20.9%. At practical brightness levels of 100 cd m$^{-2}$, we achieve $\eta_{EQE} = 17.2$% and $\eta_{EQE} = 17.8$% for the best host-free and host-guest devices, respectively.

## Results

**Synthesis and structures.** We prepared the $CF_3$-substituted carbazolate complexes **1** ($R^1 = CF_3$, $R^2 = {}^tBu$) and **2** ($R^1 = R^2 = CF_3$) (Fig. 1) from ($^{Ad}$CAAC)AuCl and the corresponding carbazoles in the presence of KO$^t$Bu, following our previously published procedures[12–14]. Complexes **1** and **2** were prepared on a 5-g scale as white solids that are stable in air and in solution for long period of time. They possess good solubility in aromatic solvents (toluene, chlorobenzene, 1,2-difluorobenzene), THF, dichloromethane, or DMF, but are insoluble in hexane. We chose the adamantyl-substituted CAAC ligand ($^{Ad}$CAAC) since its high steric hindrance provides good thermal stability and high PL intensities. Thermogravimetric analysis (TGA, 5% weight loss) showed decomposition temperatures of 325 °C for **1** and 364 °C for **2** (Supplementary Fig. 1). The data for the known[14] green ($R^1 = R^2 = H$, here complex **3**) and yellow emitters ($R^1 = R^2 = {}^tBu$, here complex **4**) are included for comparison.

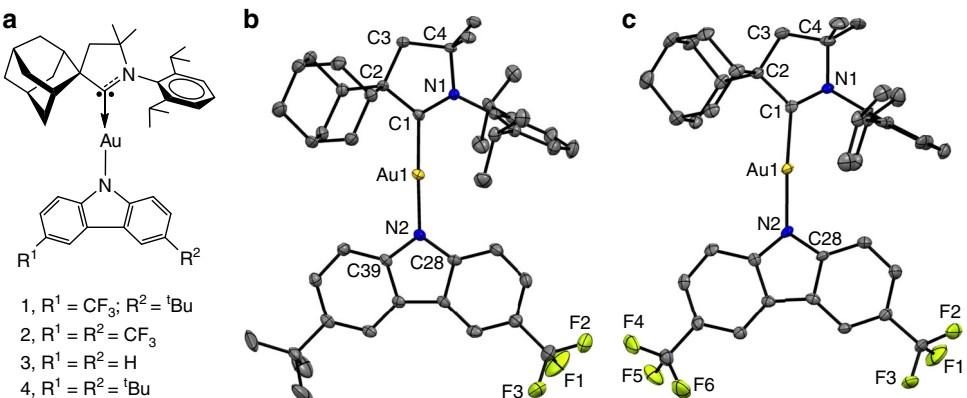

**Fig. 1 Molecular and crystal structures. a** Molecular structures of carbene–metal–amides **1**, **2**, **3** and **4**. Single-crystal X-ray structures of **1** (**b**) and **2** (**c**). Ellipsoids are shown at 50% probability.

The structures of **1** and **2** were confirmed by single-crystal X-ray diffraction (Fig. 1). Both complexes show a two-coordinate geometry for the gold atom with negligible deviation from linearity (Table 1); there are no close Au···Au contacts. The HOMO–LUMO overlap is directly related to the donor–acceptor distance C1(CAAC)···N2(Cz), which is 0.02 Å shorter in **1** than in **2**. This is likely to impact radiative rates and $\Delta E(S_1 - T_1)$[13,24]. The torsion angle N1–C1–N2–C28 for **1** of 14.3° is similar to those in **3/4**, whereas for **2** it is nearly 0°, possibly due to intermolecular interactions in the crystal (Supplementary Fig. 2).

**Electrochemical and photophysical properties**. The redox behaviour of **1** and **2** was analysed in acetonitrile solution using [$^n$Bu$_4$N]PF$_6$ as the supporting electrolyte (Supplementary Fig. 3). The electrochemical data are shown in Table 2. Both **1** and **2** show a quasi-reversible, carbene ligand-centred reduction process. Complex **1** has the smallest peak-to-peak separation ($\Delta E_p$) in the series (73 mV), indicating higher stability of the reduced species. The reduction potential is sensitive to the number of

Cz–CF$_3$ groups and leads to a greater LUMO stabilization for **1** and **2** compared with **3/4**. Both **1** and **2** show irreversible carbazole-centered oxidation processes (Supplementary Fig. 3). The HOMO levels from the onset of the first oxidation potentials[25] are −5.85 and −6.12 eV for **1** and **2**, respectively, compared to −5.61 eV for **3** and −5.47 eV for **4**.

The UV–vis absorption spectra of **1** and **2** were measured in THF (Fig. 2). All complexes show π–π* transitions at ca. 270 nm, which can be ascribed to intra-ligand (IL) transitions of the CAAC carbene, and weaker progressions around 300–310 and 360–375 nm ascribed to π–π* IL transitions of the amide. The broad, low-energy absorption band is assigned largely to ligand-to-ligand CT transitions {π(carbazole)–π* (CAAC)}. The onset of the absorption CT band and its peak position exhibit negative solvatochromism and blue-shifting by ca. 20 nm on each step of the series **4 → 3 → 1 → 2**, with decreasing electron-donor character of the amide ligand. The observed trend in absorption spectra is largely consistent with the increase of the band gap ($\Delta E$, Table 2) identified by cyclic voltammetry (CV) (see below).

On excitation with UV light ($\lambda_{exc} = 365$ nm), **1** and **2** as crystals show blue PL at $\lambda_{max} < 440$ nm, dominated by structured emission. This emission is similar to that observed for all complexes in frozen 2-MeTHF at 77 K (Supplementary Fig. 4) and can be ascribed to local (i.e. ligand-centred) triplet excited states ($^3$LE). The behaviour of **1** and **2** contrasts with that of **3** and **4** (Supplementary Fig. 5), which show unstructured CT emission in the crystalline phase. The emission red-shifts in neat amorphous thin films and in liquid solution, becoming broad and unstructured. We ascribe this emission to luminescence from CT excited states, which represent the lowest-energy triplet excitation in these less-constrained environments. The energy of the CT transition ($E_{CT}$), measured from its high-energy onset in toluene solution, increases with increasing electron-acceptor

### Table 1 Structural parameters of gold CMA complexes.

|   | Au–C1 | Au–N2 | C1···N2 | Angle C1–Au–N2 | Torsion angle N1–C1–N2–C28 |
|---|---|---|---|---|---|
| **1** | 1.983 (4) | 2.016 (3) | 3.999 (4) | 177.54 (15) | 14.3 (4) |
| **2** | 1.994 (5) | 2.024 (4) | 4.018 (4) | 176.06 (16) | 1.1 (2) |
| **3** | 1.991 (3) | 2.026 (2) | 4.017 (3) | 178.78 (11) | 17.7 (1) |
| **4** | 1.997 (3) | 2.020 (2) | 4.017 (3) | 178.25 (11) | 16.3 (1) |

Selected bond lengths [Å] and angles [°] of **1** and **2**. Measurements of **3** and **4** are reproduced for comparison (average values for the two independent molecules in the unit cell for **1** and **2**).

### Table 2 Cyclic voltammetry of gold CMA complexes.

| Complex | Reduction | | $E_{LUMO}$ | Oxidation | | | $E_{HOMO}$ | $\Delta E$ |
|---|---|---|---|---|---|---|---|---|
| | $E_{1st}$ | $E_{onset\ red}$ | eV | $E_{1st}$ | $E_{onset\ ox}$ | $E_{2nd}$ | eV | eV |
| 1 | −2.65* (73) | −2.57 | −2.82 | +0.57 | +0.46 | +1.05 | −5.85 | 3.03 |
| 2 | −2.55* (90) | −2.47 | −2.92 | +0.83 | +0.73 | – | −6.12 | 3.20 |
| 3 | −2.68* (80) | −2.60 | −2.79 | +0.26 | +0.22 | +0.77 | −5.61 | 2.82 |
| 4 | −2.86* (83) | −2.78 | −2.61 | +0.13 | +0.08 | +0.65 | −5.47 | 2.86 |

Formal electrode potentials (peak position $E_p$ for irreversible and $E_{1/2}$ for quasi-reversible processes (*), V, vs. FeCp$_2$), onset potentials ($E$, V, vs. FeCp$_2$), peak-to-peak separation in parentheses for quasi-reversible processes ($\Delta E_p$ in mV), $E_{HOMO}/E_{LUMO}$ (eV) and band gap values ($\Delta E$, eV) for the investigated complexes.
The cyclic voltammetry of complexes were measured in THF solution, recorded using a glassy carbon electrode, concentration 1.4 mM, supporting electrolyte [$^n$Bu$_4$N][PF$_6$] (0.13 M), measured at 0.1 V s$^{-1}$;
$E_{HOMO} = -(E_{onset\ ox\ Fc/Fc+} + 5.39)$ eV; $E_{LUMO} = -(E_{onset\ red\ Fc/Fc+} + 5.39)$ eV. Measurements of **3** and **4** are reproduced for comparison.

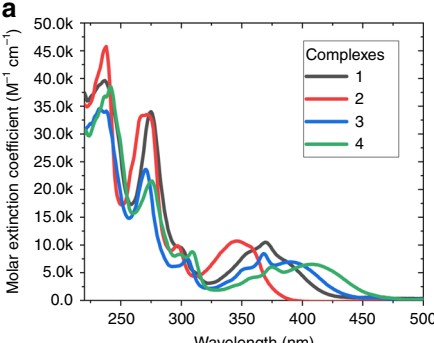
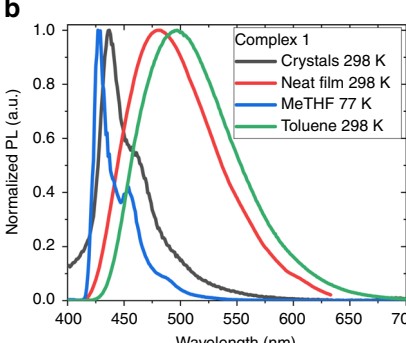
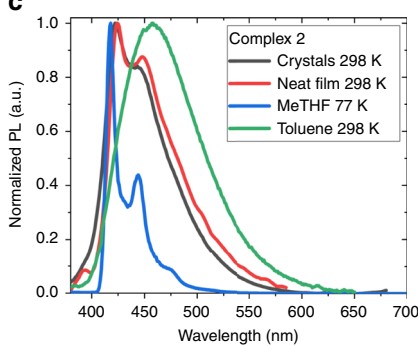

**Fig. 2 UV–vis and photoluminescence spectra of gold CMA complexes. a** UV–vis spectra in THF solution for gold complexes **1**, **2** in comparison with **3** and **4**. Photoluminescence spectra of **1** (**b**) and **2** (**c**) at 298 K as crystals, in neat film, in frozen MeTHF and in liquid toluene solution (excitation at 365 nm).

strength of the carbazole substituents, from 2.62 eV for **4** to 3.15 eV for **2**. The effect of carbazole substitution on the ³LE emission energy ($E_{LE}$), measured from its high-energy onset in frozen MeTHF, is less pronounced, with a change of only 0.11 eV between **4** and **2** (Table 3).

The CT excited-state energy and lifetime of **1** are environment dependent (Table 3). At 298 K in toluene solution, the emission lifetime is 740 ns. The peak emission energy blue-shifts by 60 meV in neat solid samples and the lifetime increases to 1 µs. In DPEPO host, which increases the peak emission energy by a further 110 meV, the lifetime increases to around 20 µs. On cooling to 77 K, the excited-state lifetimes of solid samples increase to 600–700 µs.

For **2**, excited-state lifetimes at 298 K follow a similar trend and are longer. The emission in toluene solution exhibits a featureless CT character with τ = 11.5 µs. In neat solid films, the emission shows a blue-shift with a structure associated with ³LE emission. On cooling solid samples to 77 K, lifetimes increase to over 2 ms and the ³LE character of the emission becomes dominant. The energies of the CT and ³LE states in MeTHF were measured at 298 and 77 K, respectively (Table 3). For complexes **1**, **3** and **4**, the Cz ³LE triplet state is 0.1–0.3 eV higher than the CT state, leading to negative energy gap values $\Delta E(CT-^3LE)$, which increase in the order **1** < **3** < **4**. We thus observe that emission lifetime is correlated with $E_{CT}$, increasing as the $\Delta E(CT-^3LE)$ gap narrows. ³LE phosphorescence is observed in environments where the lowest CT states are no longer the lowest-energy triplet excitations, and the PL quantum yield is reduced. For instance, the PL quantum yield in solution (298 K) drops from near unity for **1** to 61% for **2**.

**Device characterization and performance.** OLEDs utilizing complexes **1**–**4** as emitters were fabricated by thermal vapour deposition under high vacuum ($10^{-7}$ Torr) on ITO-coated glass substrates with a sheet resistance of 15 Ω/□. Two device architectures were employed, shown in Fig. 3 together with chemical structures for the materials used. Architecture A was used for the blue-emitting complexes **1** and **2**. A 40-nm layer of 1,1-bis{4-[N, N-di(4-tolyl)amino]phenyl}cyclohexane (TAPC) functions as a hole transport layer, with a 5-nm layer of 9,9′-biphenyl-2,2′-diylbis-9H-carbazole (o-CBP) acting as an exciton blocking layer due to its slightly higher $T_1$ energy and deeper HOMO. The 30-nm thick emissive layer (EML) was composed of either pure **1** and **2** in a host-free configuration, or the emitting material doped at 20 wt% into a bis[2-diphenylphosphino)phenyl]ether oxide (DPEPO) host. A 40-nm layer of diphenyl-4-triphenylsilyl-phenylphosphine oxide (TSPO1) was used as the electron-transporting and hole-blocking layer.

Devices containing the green emitter **3** and yellow emitter **4** were fabricated according to architecture B, using a 40-nm TAPC layer as a hole transport layer and a 20-nm EML utilizing **3** or **4** in a host-free configuration or doped at 20 wt.% in either DPEPO or 1,3,5-tris(carbazol-9-yl)benzene (TCP). A 10-nm layer of 1,4-bis(triphenylsilyl)benzene (UGH2) was used as a hole-blocking layer and 40 nm of 1,3,5-tris(2-N-phenylbenzimidazole-1-yl) benzene (TPBi) as an electron-transport layer.

Electroluminescence spectra for devices based on **1** are shown in Fig. 4; peak wavelengths ($\lambda_{Peak}$) and CIE colour space coordinates are summarized in Table 4. Host-free **1** devices show $\lambda_{Peak}$ = 473 nm and CIE (0.18, 0.27). The electroluminescence can be blue-shifted by dispersal in a low-polarity host material, here

**Table 3 Photophysical properties.**

| Complex | 1 | | | | 2 | | 3 | | | | 4 | | | |
|---|---|---|---|---|---|---|---|---|---|---|---|---|---|---|
| | Toluene | Neat | o-CBP:1 | DPEPO:1 | Toluene | Neat | Toluene | Neat[b] | TCP:3[b] | DPEPO:3 | Toluene | Neat | TCP:4 | DPEPO:4 |
| $\lambda_{em}$ (nm) | 495 | 484 | 479 | 464 | 456 | 425 | 528 | 500 | 500 | 485 | 552 | 540 | 535 | 514 |
| τ (µs) | 0.74 | 0.99 | 1.05 | 19.4 | 11.5 | 10.8 | 1.25 | 0.76 | 0.97 | 1.31 | 0.84 | 0.69 | 0.98 | 1.12 |
| Φ (%, 300 K; N₂) | 96 | – | – | – | 61 | – | 98 | – | – | – | 95 | – | – | – |
| $k_r$ ($10^5$ s⁻¹) | 13.1 | – | – | – | 0.53 | – | 7.8 | – | – | – | 11.3 | – | – | – |
| $k_{nr}$ ($10^5$ s⁻¹) | 0.54 | – | – | – | 0.34 | – | 0.16 | – | – | – | 0.60 | – | – | – |
| ¹CT/³LE (eV)[a] | 2.86/2.97 | | | | 3.15/3.03 | | 2.76/2.96 | | | | 2.62/2.92 | | | |
| $\Delta E(^1CT-^3LE)$ (eV)[a] | −0.11 | | | | 0.12 | | −0.20 | | | | −0.30 | | | |

Emission data of **1**, **2**, **3** and **4** in toluene solution and in thermally evaporated solid films.
[a] ¹CT and ³LE energy levels based on the onset values of the emission spectra blue edge at 77 K in MeTHF glass and in solution at 298 K.
[b] Reproduced from ref. [19].

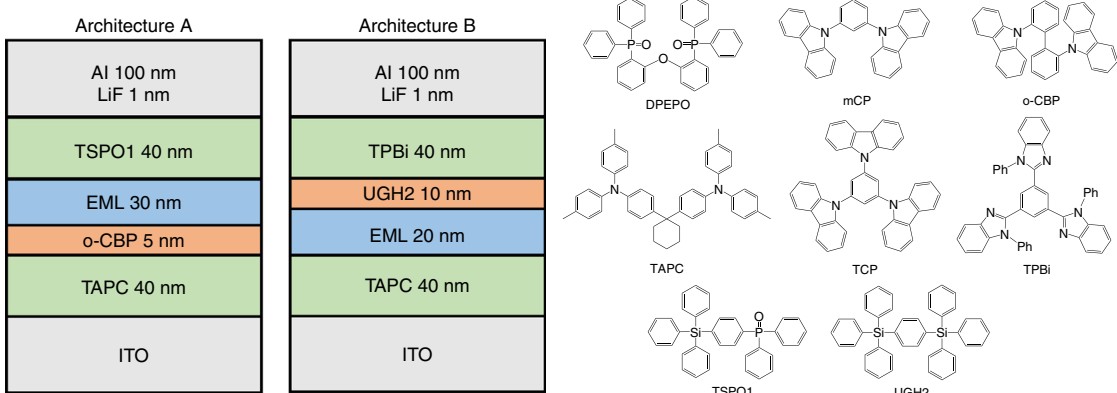

**Fig. 3 Vapour-deposited OLED device architectures.** OLED architectures A (for **1** and **2**) and B (for **3** and **4**) and chemical structures of the materials used.

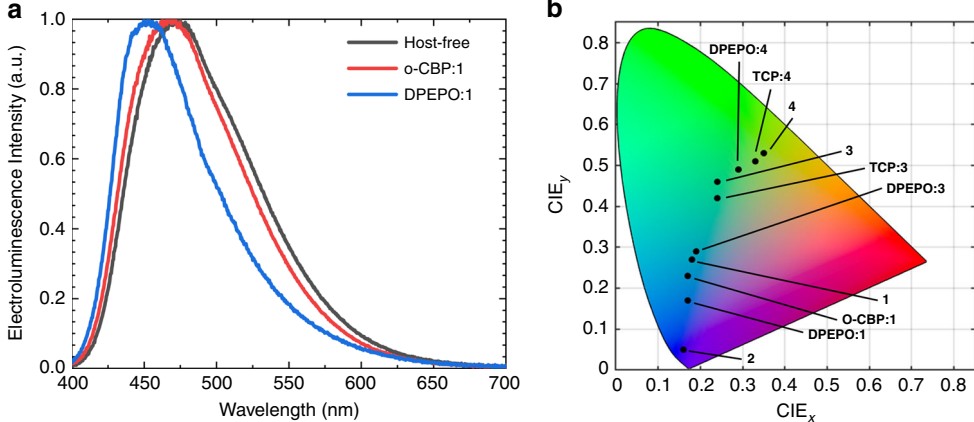

**Fig. 4 Electroluminescence curves and CIE diagram. a** Normalized electroluminescence spectra from devices incorporating **1** in host-free and host-guest structures. **b** CIE colour space chart depicting the apparent colour of electroluminescence from OLED devices incorporating CMA emitters in host-free and host-guest structures. CIE coordinates for **3** in TCP host and host-free structures are reproduced from ref. [19].

**Table 4 Performance data of evaporated OLEDs.**

| Emitting layer | $V_{On}$ [V][a] | $\eta_{EQE}$ [%] (max.) | $\eta_{EQE}$ [%] (100 cd m$^{-2}$) | $\lambda_{Peak}$ [nm] | CIE (x, y) |
|---|---|---|---|---|---|
| **1** | 3.7 | 17.3 | 17.2 | 473 | (0.18, 0.27) |
| o-CBP:**1** | 3.9 | 17.2 | 16.0 | 466 | (0.17, 0.23) |
| DPEPO:**1** | 3.7 | 20.9 | 17.8 | 450 | (0.17, 0.17) |
| **2** | – | – | – | 423 | (0.16, 0.05) |
| **3** | 3.5 | 23.1 | 23.0 | 500 | (0.24, 0.46) |
| TCP:**3** | 3.3 | 26.9 | 26.4 | 500 | (0.24, 0.42) |
| DPEPO:**3** | 4.4 | 21.9 | 20.8 | 474 | (0.19, 0.29) |
| **4** | 5.1 | 11.5 | 10.1 | 536 | (0.35, 0.53) |
| TCP: **4** | 4.7 | 18.7 | 18.4 | 527 | (0.33, 0.51) |
| DPEPO: **4** | 4.7 | 24.7 | 22.9 | 518 | (0.29, 0.49) |

Summary of champion OLED turn-on voltage, $\eta_{EQE}$ performance and spectral parameters for varying EML composition. Metrics for **3** and TCP:**3** OLEDs reproduced from ref. [19].
[a]Values at brightness >1 cd/m$^2$.

o-CBP, and further blue-shifted by utilizing a high-polarity host material, here DPEPO, to obtain a peak wavelength of 450 nm (CIE 0.17, 0.17).

Similar shifts are achievable for **3** and **4**, utilizing TCP/DPEPO as the low/high-polarity host material, respectively, in agreement with our previous report of **3** in mCP host (1,3-bis(N-carbazolyl) benzene). Host-free **2** devices show deep blue electroluminescence with $\lambda_{Peak} = 423$ nm and CIE (0.16, 0.05); however, the emission appears to be of $^3$LE character (Supplementary Fig. 6) and the devices undergo rapid degradation, with an additional spectral feature at 580 nm arising on an ~1 s timescale. Full device characterization was therefore only carried out using **1**, **3** and **4** as emitters.

Devices based on **1** show low turn-on voltages (reported as the applied bias at which luminance equals 1 cd m$^{-2}$) of $V_{On} = 3.7$ V for **1**, which indicates an absence of large barriers to charge injection into the emitting layer even in host-free architectures. Figure 5 shows the current density–voltage and luminance–voltage characteristics of champion **1** devices in both host-free and host-guest environments. We calculated EQE (Figure 6, $\eta_{EQE}$) by measuring the on-axis irradiance and assuming a Lambertian emission profile. Slightly super-Lambertian emission was previously observed for CMA-based OLEDs. We therefore consider that the $\eta_{EQE}$ estimates presented here are conservative[19]. The peak $\eta_{EQE} = 17.3\%$ for blue host-free **1** devices is an indication of the insensitivity of gold-bridged CMA materials to aggregation quenching. The efficiency rises slightly to $\eta_{EQE} = 20.9\%$ on dilution in a DPEPO host. Host-free devices exhibit reduced roll-off

compared to host-guest devices, as might be expected from a reduction in triplet density under operation. Equivalent data for compounds **3** and **4** are shown in Supplementary Figs. 7–8.

We measured the operating lifetimes of the host-free and host-guest OLED devices for complexes **1**, **3** and **4** as the time taken to reduce the initial brightness from 100 to 95 cd/m$^2$ (LT$_{95}$, Supplementary Fig. 9). The OLED devices with DPEPO as a host exhibited the shortest LT$_{95}$ values of less than 4 min. Host-free devices showed slightly longer lifetimes and devices based on **3** and **4** tended to be more stable than **1**. In the low-polarity host materials, o-CBP for complex **1** and TCP for complexes **3** and **4**, OLED devices based on complexes **1** and **4** showed only marginal improvements, whereas significant enhancement for LT$_{95}$ (2 h) was measured for complex **3**. This result can be correlated with the presence of the $^t$Bu-group for both materials **1** and **4** and instability of the cation-radical species for all complexes (Supplementary Fig. 3).

## Discussion

By varying the electron-donating or -withdrawing nature of carbazole substituents the HOMO–LUMO gap of CMA-type photoemitters can be readily adjusted and the electroluminescence colour can be tuned from yellow (CIE coordinates 0.35, 0.53) to deep blue (CIE 0.16, 0.05). In addition, the CT energy is sensitive to the host environment. These effects enable control of the energy gap between CT and LE excitations. From transient PL measurements, we establish that rapid

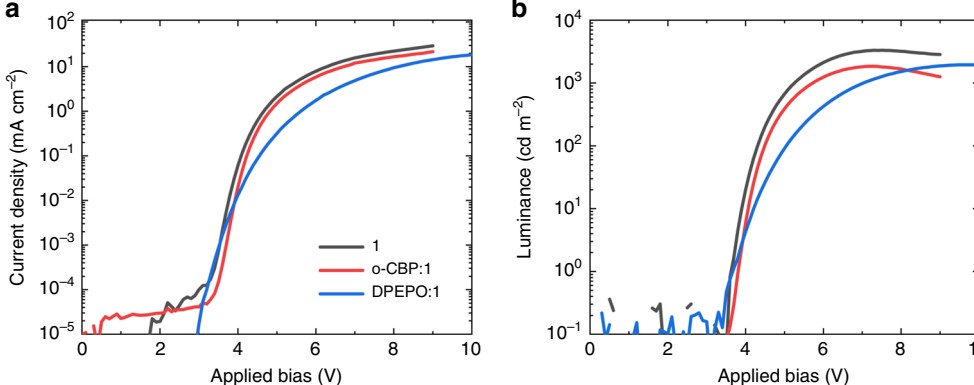

**Fig. 5 Current density–voltage–luminance (J–V–L) curves. a** Current density–voltage. **b** Luminance–voltage characteristics for OLEDs based on **1** in host-free and host-guest environments.

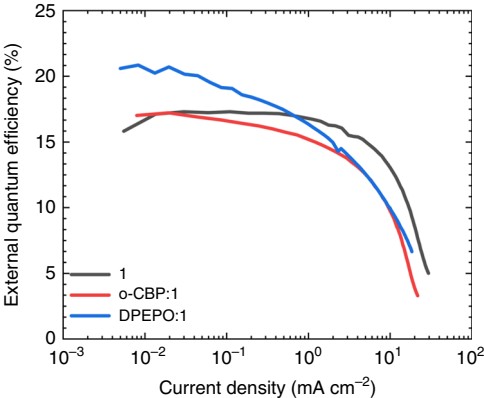

**Fig. 6 Electroluminescence external quantum efficiency.** External quantum efficiency of OLEDs based on **1** as a function of current density in host-free and host-guest environments.

triplet luminescence is correlated with a larger $\Delta E$(CT–$^3$LE) gap. Approaching resonance between the CT state and the amide triplet at around 3 eV leads to a marked reduction in performance, in contrast with the current models of all-organic TADF luminescence[26,27]. The effect is most clearly seen in the case of **2** with the most electronically depleted carbazole ligand, where in some environments the $^3$LE is the lowest-lying triplet. Where this is the case, long-lived structured phosphorescence is observed and device performance is reduced. However, the combined approach of tuning the molecular design and host environment allows devices to operate below this ceiling. We have fabricated efficient blue host-guest OLEDs with a peak $\eta_{EQE}$ of 20.9% and peak electroluminescence wavelength of 450 nm (CIE coordinates [0.17, 0.17]). We also achieved a peak $\eta_{EQE}$ of 17.3% for blue OLEDs (CIE [0.18, 0.27]) in host-free architectures. Host-free and host-guest (o-CBP) devices exhibit slow roll-off, achieving $\eta_{EQE}$ at a brightness of 100 cd m$^{-2}$ of 17.8% and 17.2%, respectively. The host-free approach can simplify the design of blue OLEDs, but the molecular design of the emitter has to be very carefully considered. We demonstrate that CMA complex **3** exhibits encouraging LT$_{95}$ values of several hours (CIE [0.24, 0.42]), while installing $^t$Bu substituents in the Cz 3,6-positions reduces LT$_{95}$ values to minutes. These findings provide a practical guide for the molecular design of the next generation of stable blue emitters for display applications.

## Methods

**General**. Solvents were distilled and dried before use. Sodium *tert*-butoxide and 3-(*tert*-butyl)phenylboronic acid, were purchased from FluoroChem; SPhos Pd G2 was purchased from Sigma-Aldrich and used as received. The carbene ligand ($^{Ad}$L)[28–30], *N*-(2-chloro-4-(trifluoromethyl)phenyl)acetamide (Supplementary Figs. 10 and 11), 6-(*tert*-butyl)-3-(trifluoromethyl)-9- acetylcarbazole (Supplementary Figs. 12 and 13) and 6-(*tert*-butyl)-3-(trifluoromethyl)-9H-carbazole (Supplementary Figs. 14 and 15)[31], and complexes ($^{Ad}$L)MCl (M = Cu and Au)[32] were obtained according to literature procedures, see Supplementary Methods. $^1$H and $^{13}$C{$^1$H} NMR spectra were recorded using a Bruker Avance DPX-300 MHz NMR spectrometer. $^1$H NMR spectra (300.13 MHz) and $^{13}$C{$^1$H} (75.47 MHz) were referenced to CD$_2$Cl$_2$ at δ 5.32 ($^{13}$C, δ 53.84) and CDCl$_3$ at δ 7.26 (δ $^{13}$C 77.16) ppm. All electrochemical experiments were performed using an Autolab PGSTAT 302N computer-controlled potentiostat. CV was performed using a three-electrode configuration consisting of either a glassy carbon macrodisk working electrode (GCE) (diameter of 3 mm; BASi, Indiana, USA) combined with a Pt wire counter electrode (99.99%; GoodFellow, Cambridge, UK) or an Ag wire pseudoreference electrode (99.99%; GoodFellow, Cambridge, UK). The GCE was polished between experiments using alumina slurry (0.3 µm), rinsed in distilled water and subjected to brief sonication to remove any adhering alumina microparticles. The metal electrodes were then dried in an oven at 100 °C to remove residual traces of water, the GCE was left to air dry and residual traces of water were removed under vacuum. The Ag wire pseudoreference electrodes were calibrated to the ferrocene/ferrocenium couple in MeCN at the end of each run to allow for any drift in potential, following IUPAC recommendations[33]. All electrochemical measurements were performed at ambient temperatures under an inert Ar atmosphere in an MeCN-containing complex under study (0.14 mM) and supporting electrolyte [n-Bu$_4$N][PF$_6$] (0.13 mM). Data were recorded with Autolab NOVA software (v. 1.11). Elemental analyses were performed by London Metropolitan University. Mass spectrometry data were obtained using APCI(ASAP) (Atmospheric Solids Analysis Probe) at the National Mass Spectrometry Facility at Swansea University.

**Synthesis of complex 1**. In a Schlenk tube, ($^{Ad}$L)AuCl (3.52 g, 5.77 mmol), 6-(*tert*-butyl)-3-(trifluoromethyl)-9H-carbazole (1.68 g, 5.77 mmol) and $^t$BuONa (0.56 g, 5.83 mmol) were stirred in THF (75 mL) for 6 h. The mixture was filtered through Celite. The filtrate was concentrated and washed with hexane to afford the product as a white solid. Yield: 93% (4.65 g, 5.37 mmol) after annealing at 140 °C under vacuum. See Supplementary Fig. 16.

$^1$H NMR (300 MHz, CD$_2$Cl$_2$): δ 8.20 (s, 1H, CH$^4$ Cz), 7.99 (d, $J$ = 2.0 Hz, 1H CH$^5$ Cz), 7.71 (t, $J$ = 7.8 Hz, 1H, *p*-CH Dipp), 7.47 (d, $J$ = 7.8 Hz, 2H, *m*-CH Dipp), 7.30 (dd, $J$ = 8.6, 2.0 Hz, 1H, CH$^7$ Cz), 7.25 (dd, $J$ = 8.6, 1.5 Hz 1H, CH$^2$ Cz), 6.87 (d, $J$ = 8.6 Hz, 1H, CH$^8$ Cz), 6.42 (d, $J$ = 8.6 Hz, 1H, CH$^1$ Cz), 4.31 (d, $J$ = 12.9 Hz, 2H, CH$_2$ Adamantyl), 2.90 (Sept, $J$ = 6.7 Hz, 2H, CH *i*Pr Dipp), 2.45 (pseudo s, 2H + 1 H, CH$_2$ CAAC overlapping with CH Adamantyl), 2.19–1.86 (m, 11H, Adamantyl), 1.44 (s, 6H, C(CH$_3$)$_2$ CAAC), 1.39 (s, 9H, *t*Bu), 1.36–1.30 (m, 12H, CH$_3$ *i*Pr Dipp). $^{13}$C NMR (75 MHz, CD$_2$Cl$_2$) δ 244.1 (s, C: CAAC), 151.9 (s, C$_q$ Cz), 148.8 (s, C$_q$ Cz), 146.2 (s, *o*-C Dipp), 140.4 (s, C–*t*Bu), 136.7 (s, *i*-C Dipp), 130.0 (s, *p*-CH Dipp), 26.8 (q, $J$ = 270.3 Hz, CF$_3$), 123.9 (s, C$_q$ Cz), 123.8 (s, C$_q$ Cz), 123.0 (s, CH$^7$ Cz), 119.8 (q, $J$ = 3.2 Hz, CH$^2$ Cz), 117.1 (q, $J$ = 31.3 Hz, C–CF$_3$ overlapping with CH$^4$ Cz), 116.9 (q, $J$ = 4.2 Hz, CH$^4$ Cz overlapping with C–CF$_3$), 115.9 (s, CH$^5$ Cz), 114.0 (s, CH$^1$ Cz), 113.9 (s, CH$^8$ Cz), 77.6 (s, s, C(CH$_3$)$_2$ CAAC), 64.5 (s, C–C: CAAC), 49.1 (s, CH$_2$ CAAC), 39.4 (s, Adamantyl), 37.6 (s, Adamantyl), 35.8 (s, Adamantyl), 34.8 (s, 2 C overlapped, C(CH$_3$)$_3$ and CH Adamantyl) 32.2 (s, C(CH$_3$)$_3$) 29.6 m, 2 C overlapped, C(CH$_3$)$_2$ and CH *i*Pr Dipp), 28.6 (s, Adamantyl), 27.7 (s, Adamantyl), 26.5 (s, CH$_3$ *i*Pr Dipp), 23.4 (s, CH$_3$ *i*Pr

Dipp). $^{19}$F NMR (282 MHz, CD$_2$Cl$_2$) δ -59.1. Anal. Calcd. for C$_{44}$H$_{54}$AuF$_3$N$_2$ (864.89): C, 61.10; H, 6.29; N, 3.24. Found: C, 61.35; H, 6.07; N, 3.43. C$_{44}$H$_{54}$AuF$_3$N$_2$H theoretical [M + H$^+$] = 865.3983, HRMS (APCI(ASAP)) = 865.3997.

**Synthesis of complex 2**. In a Schlenk tube, ($^{Ad}$L)AuCl (0.78 g, 1.28 mmol), 3,6-bis (trifluoromethyl)-9H-carbazole (0.373 g 1.28 mmol) and $^t$BuONa (0.140 mg 1.45 mmol) were stirred in THF (40 mL) for 6 h. The mixture was filtered through Celite. The filtrate was concentrated and washed with hexane to afford the product as a white solid. Yield: 85% (0.95 g, 1.08 mmol) after annealing at 140 ℃ under vacuum. See Supplementary Fig. 17.

$^1$H NMR (300 MHz, CD$_2$Cl$_2$): δ 8.27 (s, 2H, CH$^4$ Cz), 7.72 (t, $J$ = 7.8 Hz, 1H, $p$-CH Dipp), 7.48 (d, $J$ = 7.8 Hz, 2H, $m$-CH Dipp), 7.40 (d, $J$ = 8.6 Hz, 2H, CH$^2$ Cz), 6.73 (d, $J$ = 8.6 Hz, 2H, CH$^1$ Cz), 4.27 (d, $J$ = 12.7 Hz, 2H, CH$_2$ Adamantyl), 2.89 (sept, $J$ = 6.6 Hz, 2H, CH $i$Pr Dipp), 2.44 (pseudo s, 2 H + 1H, CH$_2$ CAAC overlapping with CH Adamantyl), 2.20–1.99 (m, 7H, Adamantyl), 1.97–1.85 (m, 4H, Adamantyl), 1.44 (s, 6H, C(CH$_3$)$_2$ CAAC), 1.40–1.26 (m, 12H, CH$_3$ $i$Pr Dipp). $^{13}$C NMR (75 MHz, CD$_2$Cl$_2$) δ 243.5 (s, C: CAAC), 152.3 (s, C$_q$ Cz), 146.2 (s, $o$-C Dipp), 136.7 (s, $i$-C Dipp), 130.1 (s, $p$-CH Dipp), 126.4 (q, $J$ = 270.7 Hz, CF$_3$), 125.8 (s, $m$-CH Dipp), 123.6 (s, C$_q$ Cz), 121.3 (q, $J$ = 3.3 Hz, CH$^2$ Cz), 118.8 (q, $J$ = 31.5 Hz, C–CF$_3$), 117.5 (q, $J$ = 4.1 Hz, CH$^4$ Cz), 114.7 (s, CH$^1$ Cz), 77.8 (s, C(CH$_3$)$_2$ CAAC), 64.5 (s, C–C: CAAC), 49.0 (s, CH$_2$ CAAC), 39.3 (s, Adamantyl), 37.6 (s, Adamantyl), 35.9 (s, Adamantyl), 34.7 (s, Adamantyl), 29.6 (m, 2 C overlapped, C (CH$_3$)$_2$ and CH $i$Pr Dipp), 28.6 (s, Adamantyl), 27.7 (s, Adamantyl), 26.5 (s, CH$_3$ $i$Pr Dipp), 23.3 (s, CH$_3$ $i$Pr Dipp). $^{19}$F NMR (282 MHz, CD$_2$Cl$_2$) δ -59.6. Anal. Calcd. for C$_{41}$H$_{45}$AuF$_6$N$_2$ (876.78): C, 56.17; H, 5.17; N, 3.20. Found: C, 55.83; H, 5.38; N, 3.02. C$_{41}$H$_{45}$AuF$_6$N$_2$H theoretical [M + H$^+$] = 877.3231, HRMS (APCI (ASAP)) = 877.3241.

**Photophysical characterization**. Solution UV–visible absorption spectra were recorded using a Perkin-Elmer Lambda 35 UV/vis spectrometer. UV–vis spectra of solid films were recorded using an Agilent 8453 UV–visible spectrophotometer. PL measurements for MeTHF solutions at 298 and 77 K were recorded on a Fluorolog Horiba Jobin Yvon spectrofluorometer. PL measurements for solid films were recorded using an Edinburgh Instruments FLS980 spectrometer. Photoluminescent quantum yield was measured for toluene solutions using an Edinburgh Instruments FS5 spectrometer with 350 nm excitation wavelength for complex **1** (1 mg mL$^{-1}$) and **2** (0.3 mg mL$^{-1}$), and 400 nm excitation wavelength for complexes **3** (0.5 mg mL$^{-1}$) and **4** (0.5 mg mL$^{-1}$). Toluene solutions have been prepared in a nitrogen glovebox from freshly distilled toluene and measured in a 1-cm screw-cap quartz cuvette.

**Transient PL measurements**. Time-resolved PL spectra of solid films were recorded using an electrically gated intensified charge-coupled device (ICCD) camera (Andor iStar DH740 CCI-010) connected to a calibrated grating spectrometer (Andor SR303i). Pulsed 400-nm photoexcitation was provided by second harmonic generation in a BBO crystal from the fundamental 800 nm output (pulse width = 80 fs) of a Ti:Sapphire laser system (Spectra Physics Solstice), at a repetition rate of 1 kHz. A 425-nm long-pass filter (Thorlabs) was used to prevent scattered laser signals from entering the camera. Temporal evolution of the PL emission was obtained by stepping the ICCD gate delay with respect to the excitation pulse. The minimum gate width of the ICCD was ~5 ns.

The toluene solution time-resolved fluorescence data at 298 K were collected on a time-correlated single-photon counting Fluorolog Horiba Jobin Yvon spectrofluorometer using Horiba Jobin Yvon DataStation v2.4 software. A NanoLED of 370 nm was used as excitation source, with an instrument response function width of 2 ns. The data were analysed using Horiba Jobin Yvon DAS6 v6.3 software.

The neat film and host-guest time-resolved fluorescence data at 77 K were collected on an Edinburgh Instruments FS5 spectrofluorometer using a 5-W microsecond Xe flash lamp with a repetition rate of 100 Hz (360 nm excitation wavelength).

**OLED fabrication and characterization**. OLED devices were fabricated by high-vacuum (10$^{-7}$ Torr) thermal evaporation on ITO-coated glass substrates with a sheet resistance of 15 Ω/□. Substrates were cleaned by sonication in non-ionic detergent, deionised water, acetone and isopropyl alcohol and subjected to an oxygen plasma treatment for 10 min. Layers were deposited at rates of 0.1–2 Ås$^{-1}$. o-CBP was synthesized according to the literature procedure[34]. TAPC, TCP and UGH2 were purchased from Luminescence Technology Corp. TPBi, DPEPO and TSPO1 were purchased from Shine Materials. All purchased materials were used as received. OLED current density–voltage measurements were made using a Keithley 2400 source-meter unit. The luminance was measured on-axis using a 1-cm$^2$ calibrated silicon photodiode at a distance of 15 cm from the front face of the OLED. Electroluminescence spectra were measured using a calibrated OceanOptics Flame spectrometer. Lifetime measurements were measured with a Keithley 2400 source-meter unit and a 0.75-cm$^2$ silicon photodiode. The devices were held under rough vacuum (~10$^{-3}$ Torr).

## Data availability
The data underlying this publication are available through the following web link: https://doi.org/10.17863/CAM.49735.

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

## Acknowledgements

This work was supported by the Engineering and Physical Sciences Research Council (EPSRC, Grant No. EP/M005143/1), the Royal Society, the European Research Council (ERC) and Samsung Display Corp. (SDC). P.J.C. acknowledges EPSRC for PhD studentship funding. C.S.B.M. acknowledges the Royal Society (RGF\EA\180041) and St John's College, Cambridge. M.B. is an ERC Advanced Investigator Award holder (Grant No. 338944-GOCAT). D.C. and S.T.E.J. acknowledge support from the Royal Society (grant nos. UF130278 and RG140472). A.S.R. acknowledges support from the Royal Society (Grant Nos. URF\R1\180288 and RGF\EA\181008). We thank the National Mass Spectrometry Facility at Swansea University.

## Author contributions

P.J.C. and C.S.B.M. developed and characterized the OLED devices. C.S.B.M. performed the OLED operating lifetime test. A.S.R. and F.C. performed the molecular design and synthesis. A.S.R. performed X-ray crystallography, TGA and electrochemistry. A.S.R. and F.C. performed initial steady-state photoluminescence studies. C.S.B.M. and S.J. carried out the ns–μs transient and additional steady-state photoluminescence studies. D.C., M.B. and A.S.R. planned the project and designed the experiments. P.J.C., C.S.B.M., D.C., M.B., N.C.G. and A.S.R. co-wrote the manuscript. All authors contributed to the discussion of the results, analysis of the data and reviewed the manuscript.

## Competing interests

The authors declare no competing interests.
