## [Peer Review File · Nature Communications]

REVIEWERS' COMMENTS:

Reviewer #1 (Remarks to the Author):

The authors have improved the manuscript by adding operational lifetime measurements of the CMA-based light-emitting diodes. While the stability seems rather low for the blue-emitting devices, I appreciate that these measurements have been included. For the green-emitting complex 3 published previously in Ref. 20, a fairly good operational lifetime was observed. I consider this study a good advancement in the field of CMA-type OLEDs.

Minor issue:

- While I understand that it is problematic to measure the emission profile, I would still modify the text regarding Lambertian emission while referring to Ref. 20, for correctness. One could write that Lambertian emission is assumed, although slight deviations from a Lambertian emission profile were observed for CMA-based OLEDs previously.

Reviewer #2 (Remarks to the Author):

I have had a chance to review the manuscript and I believe that the authors have adequately addressed all of the referees' comments. The quality of the manuscript has been increased as a consequence of these revisions. I would, therefore, like to recommend acceptance of the manuscript.

Response to Referees

Reviewer #1 (Remarks to the Author):

The authors have improved the manuscript by adding operational lifetime measurements of the CMA-based light-emitting diodes. While the stability seems rather low for the blue-emitting devices, I appreciate that these measurements have been included. For the green-emitting complex 3 published previously in Ref. 20, a fairly good operational lifetime was observed. I consider this study a good advancement in the field of CMA-type OLEDs.

Minor issue:

- While I understand that it is problematic to measure the emission profile, I would still modify the text regarding Lambertian emission while referring to Ref. 20, for correctness. One could write that Lambertian emission is assumed, although slight deviations from a Lambertian emission profile were observed for CMA-based OLEDs previously.

Response: We thank the reviewer for the detailed and expert review. We agree and amended the text: "We calculated external quantum efficiency (η_{EQE}) by measuring on-axis irradiance and assuming a Lambertian emission profile. Slightly super-Lambertian emission was previously observed for CMA-based OLEDs. We therefore consider that the EQE estimates presented here are conservative."

Reviewer #2 (Remarks to the Author):

I have had a chance to review the manuscript and I believe that the authors have adequately addressed all of the referees' comments. The quality of the manuscript has been increased as a consequence of these revisions. I would, therefore, like to recommend acceptance of the manuscript.

Response: We thank the reviewer for the detailed and expert review.